# An Improved Source Model of the 2021 $M_w$ 6.1 Yangbi Earthquake (Southwest China) Based on InSAR and BOI Datasets

Hao Lu [1], Guangcai Feng [1,*], Lijia He [1], Jihong Liu [1], Hua Gao [2], Yuedong Wang [1], Xiongxiao Wu [3], Yuexin Wang [1], Qi An [1] and Yingang Zhao [1]

1   School of Geosciences and Info-Physics, Central South University, Changsha 410083, China
2   School of Geography and Environment, Jiangxi Normal University, Nanchang 330022, China
3   Guangdong Land Resources Survey and Mapping Institute, Guangzhou 510599, China
*   Correspondence: fredgps@csu.edu.cn

**Abstract:** The azimuth displacement derived by pixel offset tracking (POT) or multiple aperture InSAR (MAI) measurements is usually used to characterize the north-south coseismic deformation caused by large earthquakes ($M > 6.5$), but its application in the source parameter inversion of moderate-magnitude earthquaes (~$M$ 6.0) is rare due to the insensitive observation accuracy. Conventional line-of-sight (LOS) displacements derived by the Interferometric Synthetic Aperture Radar (InSAR) have limited ability to constrain the source parameters of the earthquake with near north-south striking. On 21 May 2021, an $M_w$ 6.1 near north-south striking earthquake occurred in Yangbi County, Yunnan Province, China. In this study, we derive both the coseismic LOS displacement and the burst overlap interferometry (BOI) displacement from the Sentinel-1 data to constrain the source model of this event. We construct a single-segment fault geometry and estimate the coseismic slip distribution by inverting the derived LOS and BOI-derived azimuth displacements. Inversion results show that adding the BOI-derived azimuth displacements to source modeling can improve the resolution of the slip model by ~15% compared with using the LOS displacements only. The coseismic slip is mainly distributed 2 to 11 km deep, with a maximum slip of approximately 1.1 m. Coulomb stress calculation shows a maximum Coulomb stress increment of ~0.05 Mpa at the north-central sub-region of the Red River Fault. In addition, there is a small Coulomb stress increase at the Southern end of the Weixi-Weishan fault. The potential seismic risks on the Weixi-Weishan and Northwest section of the Red River faults should be continuously monitored.

**Keywords:** Yangbi earthquake; InSAR; BOI; coseismic slip





## 1. Introduction

At 13:48 (UTC), on 21 May 2021, an earthquake of $M_w$ 6.1 occurred in Yangbi County, Yunnan Province, China. The epicenter of the mainshock is located in the Cangshan Town, Yangbi County (25.67°N, 99.87°E), with a depth of 8 km, recorded by the China Earthquake Network Center (CENC). The United States Geological Survey (USGS) and the Harvard Global CMT catalog (GCMT) also released the focal mechanism solutions of this event (Table 1). The earthquake caused 34 casualties and damaged a lot of residential buildings and infrastructure near the epicenter area [1]. This event exhibits a typical foreshock-mainshock-aftershock sequence, showing a belt-like distribution with a northwest orientation and a nearly vertical dipping trend (Figure 1c) [2]. Furthermore, the distribution of aftershocks shows a predominantly unilateral rupture to the southeast (Figure 1b). Geological field investigation showed that the rupture of this event did not reach the surface [3], so the seismogenic fault may be a blind fault.

The source parameters of the Yangbi earthquake estimated from the teleseismic and geodetic data are shown in Table 1. Both Y. Wang et al. [4] and B. Zhang et al. [5] constructed a single fault geometry with fixed strike and dip angles. They only used the InSAR-derived

LOS displacements to invert the coseismic slip of this event. The slips estimated by the former (2~9 km) are shallower than those estimated by the latter (3~13 km). So, using only single-view InSAR observations leads to uncertainty in the slip distribution of constrained earthquakes. K. Zhang et al. [6] inverted the slip from the GPS data and reported that coseismic slips mainly concentrated at the depth of 4~12 km. However, the uncertainty of the inferred slip model is relatively large due to the sparse distribution of the GPS stations. S. Wang et al. [1] conducted joint inversion for InSAR and GNSS data and obtained coseismic slips concentrated at 2~10 km deep. Combining the GPS and regional broadband waveforms data, Chen et al. [7] derived a slip distribution deeper (2~14 km) than that derived by the aforementioned studies.

**Table 1.** Focal mechanism solutions of the 2021 $M_w$ 6.1 Yangbi earthquake.

| Source | Lon (°) | Lat (°) | Length (km) | Width (km) | Depth (km) | Strike (°) | Dip (°) | Rake (°) | Slip (m) | Depth Range (km) | $M_w$ |
|---|---|---|---|---|---|---|---|---|---|---|---|
| GCMT | 100.02 | 25.61 | - | - | 15.0 | 315 | 86 | 168 | - | - | 6.1 |
| USGS | 100.012 | 25.765 | - | - | 9.0 | 135 | 82 | −165 | - | - | 6.1 |
| CENC | 99.87 | 25.67 | - | - | 8.0 | 138 | 81 | −160 | - | - | $M_s$ 6.4 |
| Y. Wang et al. [4] | 99.932 | 25.646 | 14.0 | 3.0 | 2.25 | 138.8 | 87.2 | - | 0.9 | 2~9 | 6.06 |
| B. Zhang et al. [5] | - | - | 10.9 | 1.9 | 7 | 315 | 86 | - | 0.61 | 3~13 | 6.14 |
| K. Zhang et al. [6] | - | - | 28.0 | - | - | 135.0 | 80 | - | 0.8 | 4~12 | 6.04 |
| S. Wang et al. [1] | 99.91 | 25.65 | 20.0 | 8.0 | 4.92 | 134.88 | 80 | −170 | 0.8 | 2~10 | 6.07 |
| Chen et al. [7] | 99.88 | 25.66 | 18.0 | - | 8.0 | 138 | 80 | −159 | 0.95 | 2~14 | 6.10 |
| This study | 99.891 | 25.685 | 13.1 | 1.42 | 4.14 | 314 | 86.65 | 167 | 1.1 | 2~11 | 6.11 |

Differential InSAR (D-InSAR) technique can only measure the LOS displacement at the surface, which is insensitive to the N-S component. For strike-slip faulting with a predominant N-S component, the InSAR-derived observations have weak constraint on the N-S component, leading to large uncertainties in the kinematic inversion results [8,9]. Azimuth displacements derived by burst overlap interferometry (BOI) provide good constraints on the N-S component and compensate for the insensitivity of InSAR-derived LOS displacements in this direction. The pixel offset-tracking (POT) and multiple aperture InSAR (MAI) techniques can also measure the azimuth displacements, but they have lower monitoring accuracy (a few meters) than the BOI technique [8,10–12]. In addition, they cannot obtain coseismic azimuth displacement with a high signal-to-noise ratio (SNR) for small and medium-sized earthquakes, such as the Yangbi event [11,12]. The observation data derived by BOI provide important constraints on the displacement pattern in the azimuth direction, but their role in slip distribution inversion remains unclear. Therefore, it is necessary to re-estimate the fault geometry and coseismic slip distribution of the Yangbi earthquake by adding near-field BOI-derived azimuth displacements.

In this study, we first use the D-InSAR and BOI measurements from Sentinel-1 SAR data to obtain the complete coseismic displacement fields of the 2021 Yangbi earthquake. We also measure the spatiotemporal evolution of the postseismic deformation within the first 9 months after the earthquake using the Sentinel-1 data and analyze the spatial and temporal relationships between the coseismic and postseismic displacements. Then, we estimate the fault geometry and coseismic slip distribution of this event by jointly inverting the intermediate-field InSAR-derived LOS and near-field BOI-derived azimuth displacements. Next, we calculate the static stress changes on surrounding active faults by the preferred coseismic slip model. Finally, we discuss the effect on coseismic slip under the constraint of BOI-derived azimuth observation data and the regional potential seismic risks.

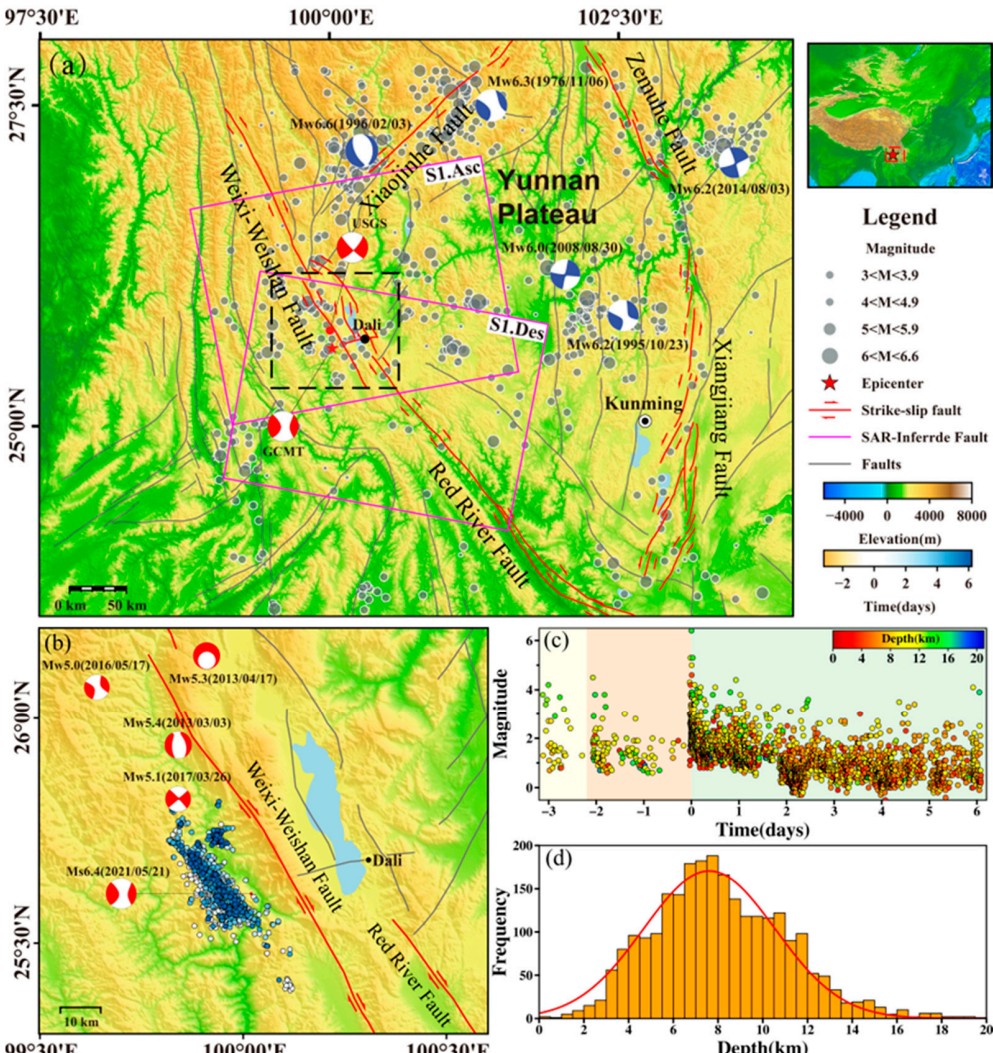

**Figure 1.** Tectonic setting of the seismogenic area of the 2021 Yangbi earthquake. (**a**) Coverage of the Sentinel-1 SAR data in the study area. Red dot and star represent the epicenters given by USGS and GCMT, respectively. Red beach balls show the focal mechanism solutions (FMS). Gray circles display the distribution of historical earthquakes (*M* > 3.0) from GCMT since 1976. Blue beach balls denote the FMS of *M* > 6.0 historical earthquakes. Thin grey lines and thick red lines are the regional active faults and the typical strike-slip faults, respectively [13]. Magenta boxes show the spatial frames of the Sentinel-1 SAR data on ascending and descending tracks. Black dotted box shows the region in (**b**). (**b**) Topographic and active faults map surrounding the Yangbi event. Red beach balls represent the FMS and epicenter locations of *M* > 5.0 earthquakes since 1976. The distributions of the shocks occurred three days before and six days after the mainshock are shown by color-coded circles. (**c**) the magnitude-time evolution of the fore- and after-shock sequence by depth-dependent color-coded circles. (**d**) the depth frequency of the fore- and after-shocks.

## 2. Tectonic Setting and Regional Seismicity

The internal blocks of the Qinghai-Tibet Plateau are pushed to the east under the continuous collision of the Indian and Eurasian plates but are blocked by the Sichuan Basin and cause tectonic movements with clockwise rotation [14]. This long-term relative movement trend has led to strong regional tectonic movements and formed active fault zones, such as Xiaojiang, Red River, and Zemuhe faults. They jointly regulate the crustal seismicity of the Sichuan-Yunnan block (Figure 1a) [15]. The Yangbi earthquake is located near the Weixi-Weishan fault (WX-WSF) and Red River fault (RRF). These two NW-trending

faults separate the Sichuan-Yunnan block from the Southwest Yunnan block. They are important for understanding the tectonic background of this earthquake [16].

The NNW-trending WX-WSF connects the Jinshajiang fault to the north and the RRF to the south, with a total length of approximately 280 km. It is dominated by a right-lateral strike-slip motion with a normal component. Since the late Pleistocene, the horizontal slip rate along the WX-WSF has been 1.8~2.4 mm/yr [17]. The RRF located in the Southeast of the seismogenic area has gradually evolved from early sinistral motion to dextral motion [18]. GPS observations showed that the average slip rate of the RRF is approximately 4.9 mm/yr [19]. Considering the geological and seismicity features, Guo et al. [20] divided the RRF into three sections, the Northern, Central and Southern sections, and their slip rates inverted by Lu et al. [21] are approximately 4.7 mm/yr, 2.3 mm/yr, and 3.6 mm/yr, respectively.

Local earthquake records show a strong seismicity between the RRF and the WX-WSF. Four moderate-magnitude earthquakes ($M > 5.0$) recorded by GCMT catalog occurred on the WX-WSF during the past decade (Figure 1). These earthquakes occurred on secondary or blind faults near the RRF, and formed earthquake sequences or swarms. The interseismic tectonic stress accumulated in the northern section of the RRF Zone, so the regional tectonic motion in this section is stronger than in the Central and Southern sections [21]. The seismicity in the Southern and Central parts of the RRF is low, which may be related to the Southeastward movement of the Sichuan-Yunnan block [22].

## 3. Data Processing

### 3.1. InSAR Measurements

In order to obtain the coseismic deformation field of the $M_w$ 6.1 Yangbi earthquake, we process 7 Sentinel-1A/B ascending images on track 99 and 7 descending images on track 135 for 5 pre- and 2 post-earthquakes. The image spatial coverage is shown in Figure 1a, and the data information is listed in Table 2. We construct a spatial-temporal baseline network for SAR images (Figure S1) [23]. We use the GAMMA software to process the SAR data by D-InSAR technique [24]. A multi-looking operation of 20 × 4 (range × azimuth) is used to improve the SNR of the differential interferograms. The 1-arc-second (~30 m resolution) Shuttle Radar Topography Mission (SRTM) digital elevation model (DEM) is used to correct the topographic phase component. A modified Goldstein filtering method is used to filter the interferograms [25]. The minimum cost flow method (MCF) is used to unwrap the filtered interferograms, and the unwrapped interferograms are geocoded into the WGS-84 coordinate system [26]. Furthermore, in order to eliminate the incoherent and low-quality points in the zones with water and dense vegetation, the coherence and amplitude thresholds are set as 0.4 and 0.3, respectively. We mask the near-field deformation regions in all interferograms and use a polynomial fitting method to remove the orbital errors and topography-related atmospheric errors. Then, we calculate the standard deviation (STD) for multiple sets of masked deformation results (Figures S2 and S3). The coseismic deformation field that are slightly affected by temporal baseline, the STD and integrity of deformation field is selected as the optimal result (Figure 2). The finally selected pre- and post-earthquake image information is listed in Table 2 (shown in bold).

**Table 2.** Parameters of the SAR images used in this study.

| Satellite | Orbits | Acquisition Dates D-InSAR | Number of Images | Acquisition Dates SBAS-InSAR | Number of Images |
|---|---|---|---|---|---|
| Sentinel-1 | Ascending (T99) | Before the earthquake 25 February 2021 2 April 2021 14 April 2021 8 May 2021 **20 May 2021** After the earthquake 26 May 2021 **1 June 2021** | 10 | Post-seismic deformation 1 June 2021–20 February 2022 | 20 |
| | Descending (T135) | Before the earthquake 23 March 2021 4 April 2021 16 April 2021 28 April 2021 **10 May 2021** After the earthquake **22 May 2021** 3 June 2021 | 10 | Post-seismic deformation 1 June 2021–20 February 2022 | 19 |

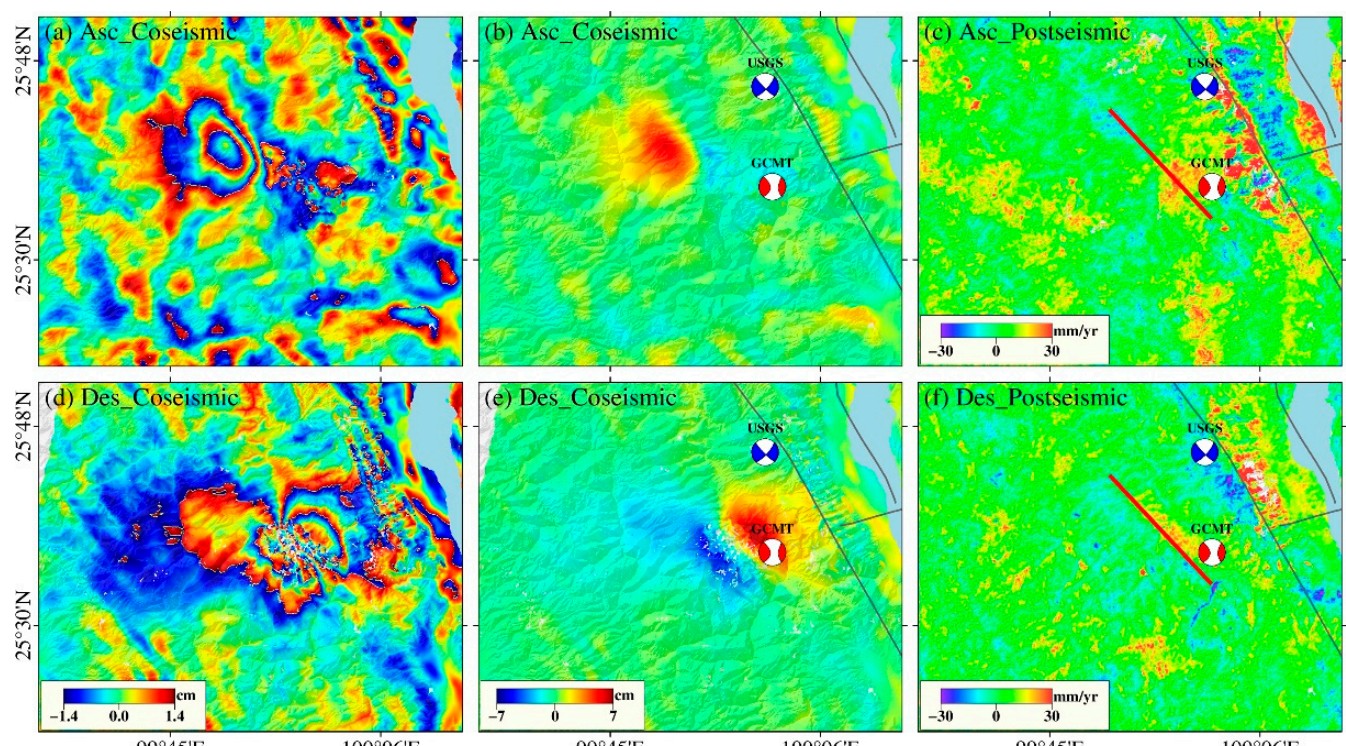

**Figure 2.** Coseismic and postseismic deformation fields of the 2021 Yangbi earthquake. (**a**,**d**) are the ascending and descending InSAR-derived interferograms, respectively. (**b**,**e**) are the corresponding unwrapped LOS deformations. (**c**,**f**) are the same as (**a**,**d**) but for the postseismic deformation rate. Red and blue beach balls show the focal mechanism solutions supplied by the USGS and GCMT catalogs, respectively. Red lines in (**c**,**f**) are the seismogenic faults used in source modeling. Thin grey lines indicate the regional active faults.

In order to obtain the postseismic deformation, we process the Sentinel-1A images acquired in the first 9 months after the earthquake using a modified small baseline subset InSAR (SBAS-InSAR) method [27,28]. This method can obtain highly coherent interferograms by combining various short baselines (image acquisition is shown in Table 2). Firstly, we generate the differential interferometric pairs with a multi-look ratio of 20:4 (range: azimuth). Then, we use the intensity map to calculate the amplitude dispersion of each pixel and set an intensity threshold of 0.3 to remove the pixels in water areas. We use the coherence of interferogram to calculate the average coherence of each pixel, and set the average coherence threshold of 0.4 to remove low quality points. Notably, only the pixel

points within the corresponding thresholds are used for time series InSAR analysis. After removing the linear deformation component, the original time series displacements obtained by the singular value decomposition (SVD) method contain nonlinear deformation, atmospheric delay and phase noise components. We eliminate the atmospheric delay by temporal high pass filtering and spatial low pass filtering. Finally, the post-earthquake deformation rates of the 2021 Yangbi earthquake are shown in Figure 2c,f.

### 3.2. 2.5-D Displacement Determination

The relationship between the one-dimensional (1-D) LOS/azimuth deformation and the three-dimensional (3-D) deformation can be expressed as follows:

$$\begin{cases} d_{LOS} = -d_e\cos(\alpha)\sin(\theta) + d_n\sin(\alpha)\sin(\theta) + d_v\cos(\theta) \\ d_{AZI} = d_e\sin(\alpha) + d_n\cos(\alpha) \end{cases} \tag{1}$$

where $d_{LOS}$ is the LOS deformation. $d_{AZI}$ is the azimuth deformation. $d_e$, $d_n$ and $d_v$ are the E–W, N–S and vertical components of the 3-D deformation, respectively. $\alpha$ and $\theta$ are the azimuth and incidence angles of the SAR satellite, respectively.

As the seismogenic fault of the Yangbi earthquake is close to N–S striking, it is challenging to calculate the 3-D deformation using only the ascending and descending LOS observations. Here, we ignore the N–S displacements and calculate the 2.5-dimensional (2.5-D) coseismic deformation by the least squares method using the combination of the ascending and descending LOS deformation [29]. The calculation formula is

$$\begin{cases} d_{LOS,ASC} = -d_e\cos(\alpha_A)\sin(\theta_A) + d_v\cos(\theta_A) \\ d_{LOS,DES} = -d_e\cos(\alpha_D)\sin(\theta_D) + d_v\cos(\theta_D) \end{cases} \tag{2}$$

We use Equation (1) to calculate the E–W and vertical components of the overlap regions, combining InSAR-derived LOS and BOI-derived azimuth deformation. The E–W and vertical components of the 2.5-D deformation of the 2021 Yangbi event are shown in Figure 3.

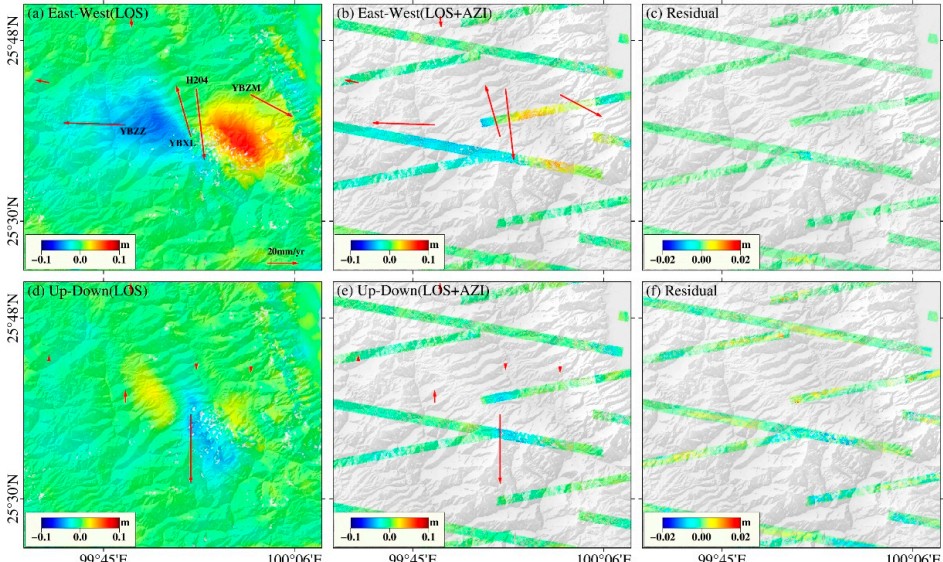

**Figure 3.** (**a**) East–west (E–W) and (**b**) vertical components of the 2021 Yangbi earthquake calculated from LOS deformation. (**b**) E–W and (**e**) vertical components calculated from LOS and azimuth deformation. Red arrows denote the GPS horizontal and vertical displacement vectors. (**c,f**) Difference between (**a,b,d,e**), respectively.

### 3.3. BOI Measurements

The D-InSAR technique can only capture the deformation in the LOS direction. The POT and MAI techniques can measure the ground displacement in the azimuth direction, but they cannot obtain the azimuth coseismic deformation fields of the 2021 Yangbi earthquake with high SNR because this event did not rupture the surface. Here, we use the BOI technique to measure the azimuth displacements.

The Sentinel-1 satellite has a standard data acquisition mode, the Terrain Observation by Progressive Scan (TOPS). The satellite has many bursts in the azimuth direction that form an overlap region as wide as 1.5 km. In the overlap region, the SAR data obtained from two adjacent bursts are considered as the backward- and forward-looking SLC images. The main steps of the BOI method include generation and processing of backward- and forward-looking interferograms and the differential processing of these two interferograms. However, it is difficult to achieve accurate co-registration between external DEM data and the SAR data in the overlap region. We geocode the entire SAR data region and extract the geographic coordinate of these overlap regions. Based on the acquisition time of the images in the overlap region, we extract four SLC images in a target overlap region from a pair of co-registration S1 data. We obtain the corresponding interferograms by conjugate multiplication of the two backward SLCs and the two forward SLCs separately. By doing so, we obtain the azimuth displacement phase [12,30]. We set the multi-look ratio of 20:4 (range: azimuth). We use the same DEM and filtering method to remove the topographic phase and noise phase, respectively. The coseismic azimuth deformation derived by the BOI technique is shown in Figure 4.

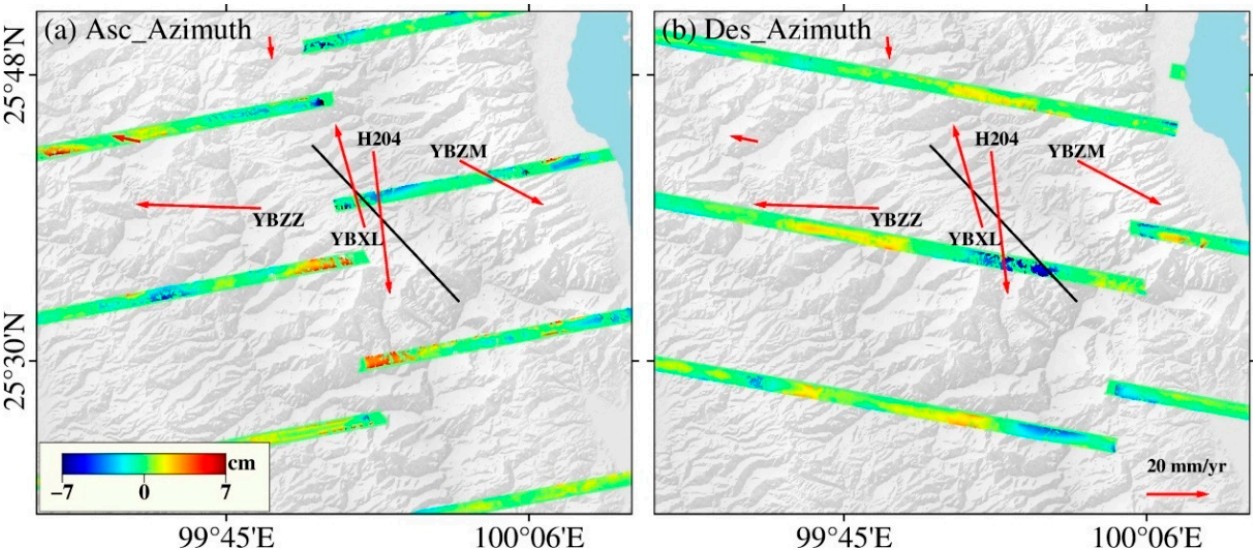

**Figure 4.** (**a**) Ascending and (**b**) descending BOI-derived azimuth displacements of the 2021 Yangbi earthquake. Red arrows indicate the GPS horizontal deformation vectors. The black line is the seismogenic fault used in source modeling.

### 3.4. Results

3.4.1. Coseismic Displacements

In Figure 2, the ascending and descending InSAR-derived interferograms have different deformation patterns in the epicentral region. The ascending results show that the major deformation is located in the Northwest of the epicenter, with a maximum of approximately 5.9 cm. The descending results show two symmetrical deformation fringes in the north-east direction, with a maximum value of approximately 6.8 cm. In order to compare the InSAR-derived LOS displacements with the GPS data, we project the 3-D displacements of the GPS data into the LOS direction according to Formula (1). We calculate the root mean square error (RMSE) between the ascending LOS displacements and the corresponding GPS projected displacements, which is 0.64 cm. Similarly, the RMSE between the descend-

ing LOS and GPS displacements is 0.76 cm. The ascending and descending BOI-derived azimuth displacements show different patterns in both sides of the seismogenic fault. The maximum ascending and descending displacement values are approximately 12.2 cm and 9.7 cm, respectively.

### 3.4.2. Postseismic Displacements

The InSAR-derived postseismic deformation is approximately 3.5 cm/yr (Figure 2c,f), and it mainly occurred in the Southeast segment of the seismogenic fault, where most aftershocks occurred. The descending postseismic deformation shows an uplift trend on the East of the seismogenic fault, similar to the coseismic deformation, and the maximum cumulative deformation is approximately 18 mm (Figure 2f). The ascending postseismic deformation shows an uplift of approximately 15 mm in the Southeast segment of the seismogenic fault (Figure 2c), which is different from the subsiding pattern of the ascending coseismic deformation in this region (Figure 2b).

## 4. Source Modeling

Based on the coseismic deformation derived from the D-InSAR and BOI measurements, we used a uniform elastic half-space model [31] to invert the fault geometry and slip distribution of the 2021 Yangbi event. In order to improve the computational efficiency of the inversion, we use a saliency-based quadtree sampling algorithm [32] to downsample the InSAR-derived LOS displacements (Figure 5a,b). Before data downsampling, we manually mask the water and low coherence areas to reduce the influence of atmospheric delay and unwrapping errors on model inversion. We perform uniform downsampling for the BOI-derived azimuth displacements. We determine the covariance function of each dataset and construct the variance-covariance matrix to give weight to the downsampled datasets.

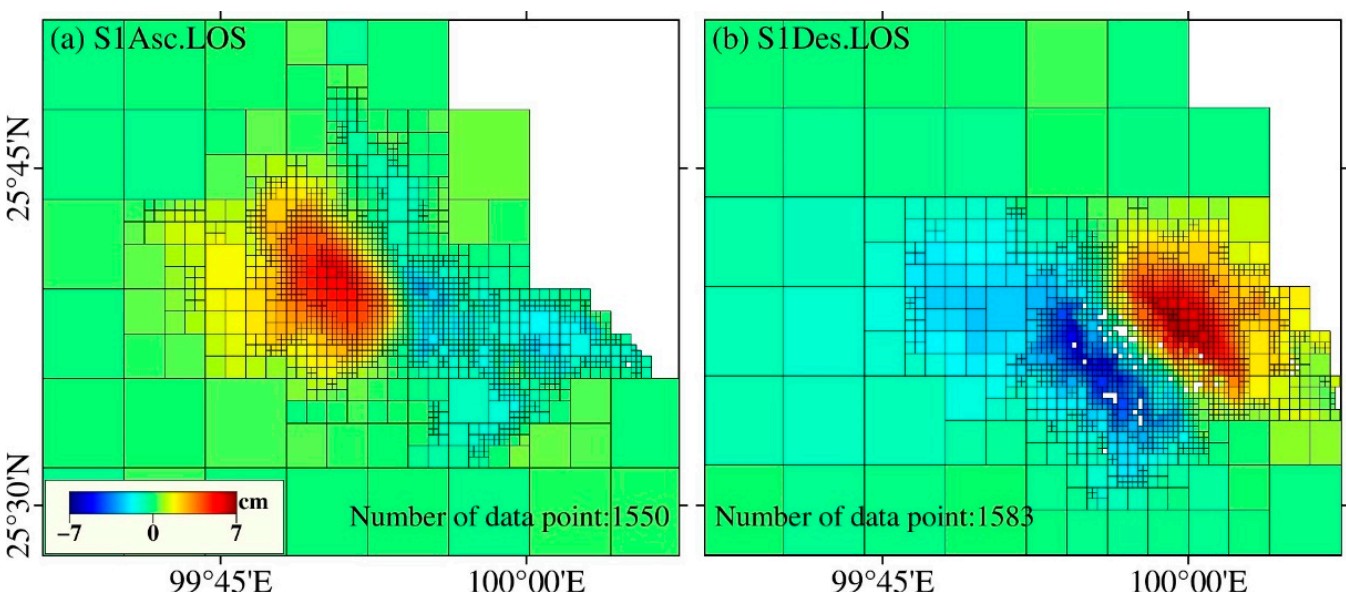

**Figure 5.** Downsampling of the (**a**) ascending and (**b**) descending InSAR-derived LOS coseismic displacements of the 2021 Yangbi earthquake.

### 4.1. Uniform Slip Inversion

The field survey shows that the 2021 Yangbi earthquake did not show obvious surface ruptures [3]. Therefore, the location of the seismogenic fault cannot be directly extracted from the geodetic observations or geological survey data. According to the relocated foreshock and aftershock sequences (Figure 1b), the seismogenic fault has a NW-striking and it is located in the Southwest of the WX-WSF [2]. We use the Bayesian method to determine the fault geometry parameters of the seismogenic fault [33].

We perform a nonlinear search for the 9 unknown parameters of the seismogenic fault based on the GBIS open-source software [34]. We use a Markov Chain Monte Carlo (MCMC) method [35,36] to estimate the fault geometry parameters and the corresponding uncertainty. Based on the FMS provided by the USGS, we constrain the fault strike to vary between $270°$ and $360°$ and the dip angle to vary between $0°$ and $90°$. We perform $10^6$ iterations to sample the posterior probability density function (PDFs). The first $5 \times 10^4$ iterations are not retained as they represent the burn-in period/step size adjustment. The inferred model parameters and their uncertainties are listed in Table 3.

**Table 3.** The inferred fault geometry parameters of the 2021 Yangbi earthquake.

| Parameters | Length (km) | Width (km) | Depth (km) | Dip (°) | Strike (°) | X Center (km) | Y Center (km) | Strike-Slip (m) | Dip-Slip (m) |
|---|---|---|---|---|---|---|---|---|---|
| Optimal | 13.10 | 1.42 | 4.14 | 86.65 | 314 | −9.43 | 3.77 | 1.99 | 0.08 |
| Mean | 12.93 | 1.64 | 4.03 | 86.80 | 314 | −9.51 | 3.82 | 1.76 | 0.07 |
| Median | 12.93 | 1.59 | 4.03 | 86.82 | 314 | −9.50 | 3.82 | 1.79 | 0.07 |
| 2.5% | 12.09 | 1.37 | 3.77 | 89.15 | 313 | −9.86 | 3.51 | 1.31 | 0.03 |
| 97.5% | 13.82 | 2.20 | 4.25 | 84.34 | 315 | −9.17 | 4.12 | 1.99 | 0.12 |

Notes: X center and Y center represent the coordinates of the midpoint of the edge associated with the reference point (25.61°N, 100.02°E) in the local coordinate system.

Figure 6 shows the resulting histograms of the marginal posterior PDFs distributions for the inferred parameters. The 1-D posterior distribution of the fault length shows a high probability at 13 km, which is smaller than that (~20 km) determined by the aftershock distribution. The seismogenic fault dips $87°$ toward the Southwest, in agreement with that given by GCMT and the inversion results of [4].

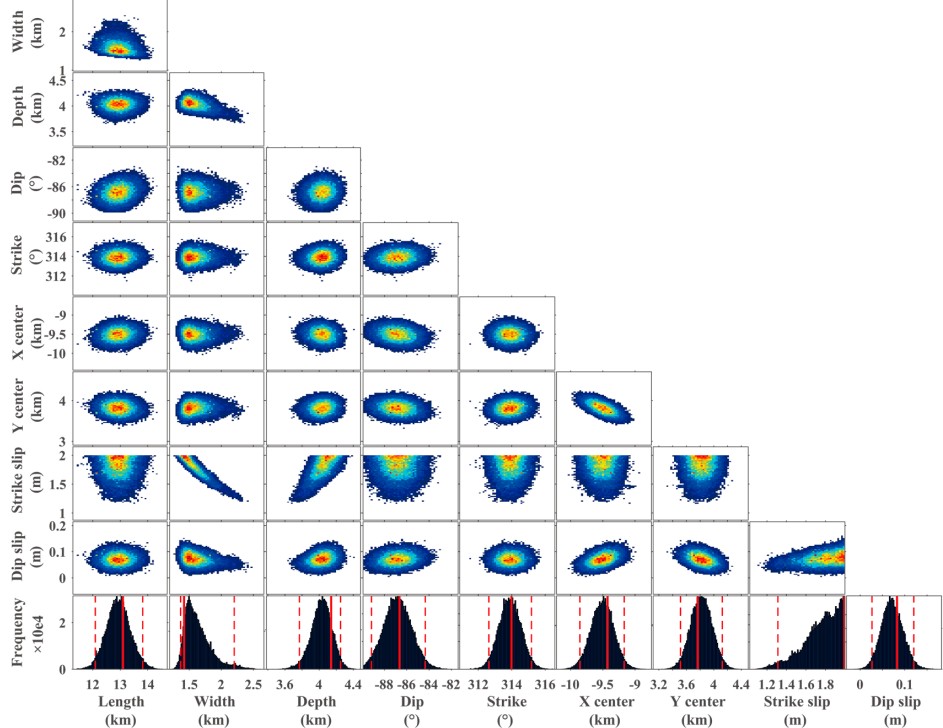

**Figure 6.** 1-D and 2-D posterior PDF plots of the fault geometry parameters of the 2021 Yangbi earthquake. The bottom row is the histograms of the marginal probability density distribution for each parameter. Red solid lines represent the maximum a posteriori probability solution and red dashed lines denote the 95% confidence interval bounds.

*4.2. Finite Fault Slip Model*

In order to invert a very precise coseismic slip distribution of the 2021 Yangbi earthquake and to test the contribution of BOI-derived azimuth displacements for the source model, we built two models: one is constrained by only the InSAR-derived LOS displacements (hereafter named model 1), and the other is jointly constrained by the InSAR-derived LOS and BOI-derived azimuth displacements (hereafter named model 2).

In order to reduce the boundary effect on the inversion results, we extend the fault length and width to 23 km and 14 km, respectively, and divide the fault plane into 1 km × 1 km rectangular patches. To calculate the Green's function, we compute the displacements caused by unit strike-slip or dip-slip on rectangular dislocation elements in a uniform elastic half-space domain, assuming a Poisson ratio of 0.25 and a rigidity of 30 GPa. To avoid abrupt changes of fault-slip between neighboring patches, we impose the second-order Laplace smoothing constraint [37]. We utilize the fast non-negativity constrained least squares algorithm to invert the coseismic slip distribution [38].

To select an appropriate smoothing factor for the coseismic slip inversion, we test 50 smoothing factors, which are logarithmically scored from 0.01 to 10 (Figure S4). When a small smoothing factor (0.01) is used, the slip distribution is highly oscillatory, which is unlikely to be plausible (Figure S5a). When a large smoothing factor (0.35) is used, the fault slip is too smooth (Figure S5b). A small smoothing factor may lead to a severe mismatch, but a large smoothing factor does not greatly improve the misfit. Therefore, we choose 0.05 as the smoothing factor value by analyzing the trade-off curve between RMS misfit and model roughness (Figure S5c).

Figure 7 shows the coseismic slip distribution models for model 1 (Figure 7a) and model 2 (Figure 7b). Both models indicate that this event is dominated by a dextral strike-slip with normal fault component, with the southwest dip angle of 87°, consistent with the spatial distribution characteristics of aftershocks. The coseismic slip distribute along the main rupture direction at the depth of 2~11 km. The slip of Model 1 in the shallow parts indicate that no significant surface rupture occurred. Model 2 has a significant slip increase in the fault depth range of 6~11 km. We also found that the slip value at the shallow part of the fault decreases after adding the BOI-derived azimuth deformation in model 2. In addition, we superimpose the location of BOI deformation points used on the differences of those two models (Figure 7d). We found obviously spatial correlation between the slip variations of the two models and location of BOI points. Although the azimuth deformation obtained by BOI only appears in the burst overlap region, its contribution to the model 2 is obvious (Figure 7d). The two models show the maximum slips of 1.13 m and 1.1 m at ~6 km depth, and moment magnitudes of $M_w$ 6.11 and $M_w$ 6.12, respectively, similar to those given by USGS and GCMT (Table 1).

In addition, we project the aftershock locations on the coseismic slip distribution map obtained by model 2 (Figure 7c). We find that the source depth of most aftershocks is deeper than 6 km. The mainshock slips is mainly concentrated approximately 6 km deep and show a complementary distribution with aftershocks on the whole fault surface [39]. This also demonstrates that BOI-derived azimuth displacements provide effective constraints on model inversion.

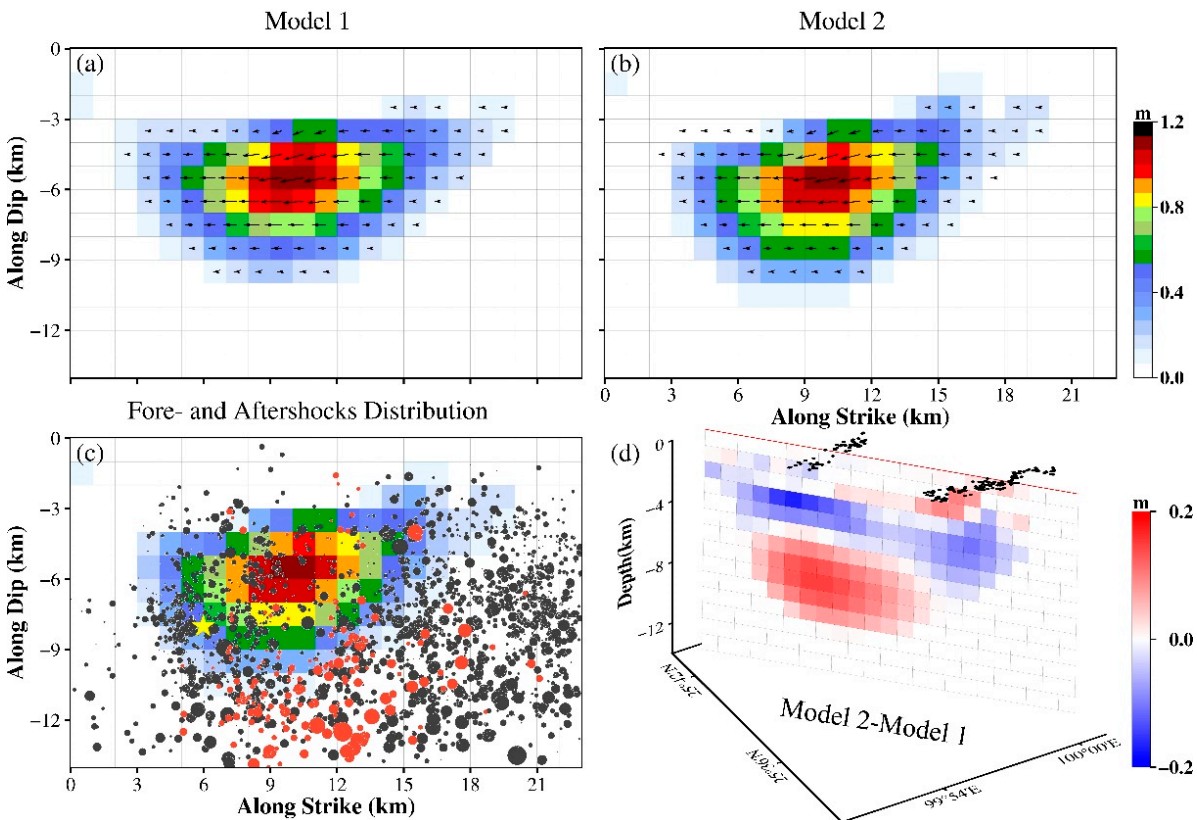

**Figure 7.** Slip distribution of the Yangbi earthquake derived from Sentinel-1 data. (**a**) is derived from InSAR only and (**b**) is derived jointly from InSAR and BOI. The black arrows in (**a**,**b**) indicate the slip direction. (**c**) Relationships between shocks distribution and coseismic slip distribution of Model 2. The orange and gray-black dots denote the relocated foreshocks and aftershocks on the fault plane, respectively [2]. The yellow star represents the position of the main earthquake. (**d**) Slip difference between the two models, with black dots showing the azimuth deformation points of the overlying faults obtained by BOI.

We use model 1 and model 2 to simulate the LOS and BOI-derived azimuth deformation of the ascending and descending tracks separately (Figures 8 and 9). Both models fit well the coseismic LOS deformation characteristics. By model 1, the RMSEs of the fitted residuals for the ascending and descending tracks are 7.4 mm and 7.6 mm, respectively. By model 2, the correspondences are 7.1 mm and 7.5 mm, slightly smaller than those of model 1. We also compare the differences between the BOI observations and the BOI simulated by model 1 in Figure 9. They are significantly different. The calculated RMSEs of the fitted residuals for the ascending and descending tracks are 25.1 mm and 23.9 mm, respectively. On the contrary, model 2 fit the BOI-derived azimuth deformation well (Figure 9). The correspondence RMSEs of the fitted residuals are 10.5 mm and 10.7 mm. This discrepancy means that the model without BOI has less constraint ability for fitting a new dataset. Although adding BOI-derived azimuth displacements does not significantly reduce the fitted LOS residuals of the model, it provides more near-field azimuth deformation useful for obtaining more detailed slip distribution.

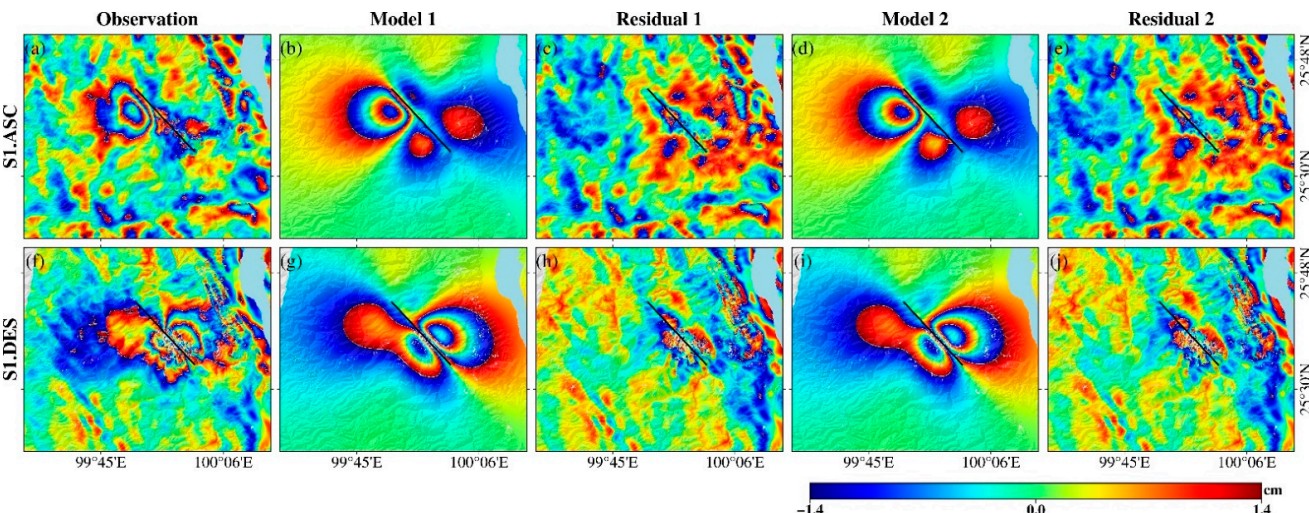

**Figure 8.** Observed and simulated LOS coseismic deformation fields and residuals obtained by Model 1 and 2 from the ascending data (**upper row**) and descending data (**lower row**). (**a**,**f**) The observation fields. (**b**,**g**) The modelled displacement fields form Model 1. (**d**, **i**) The modelled displacement fields form Model 2. (**c**,**h**,**e**,**j**) The residuals from Model 1 and 2, respectively.

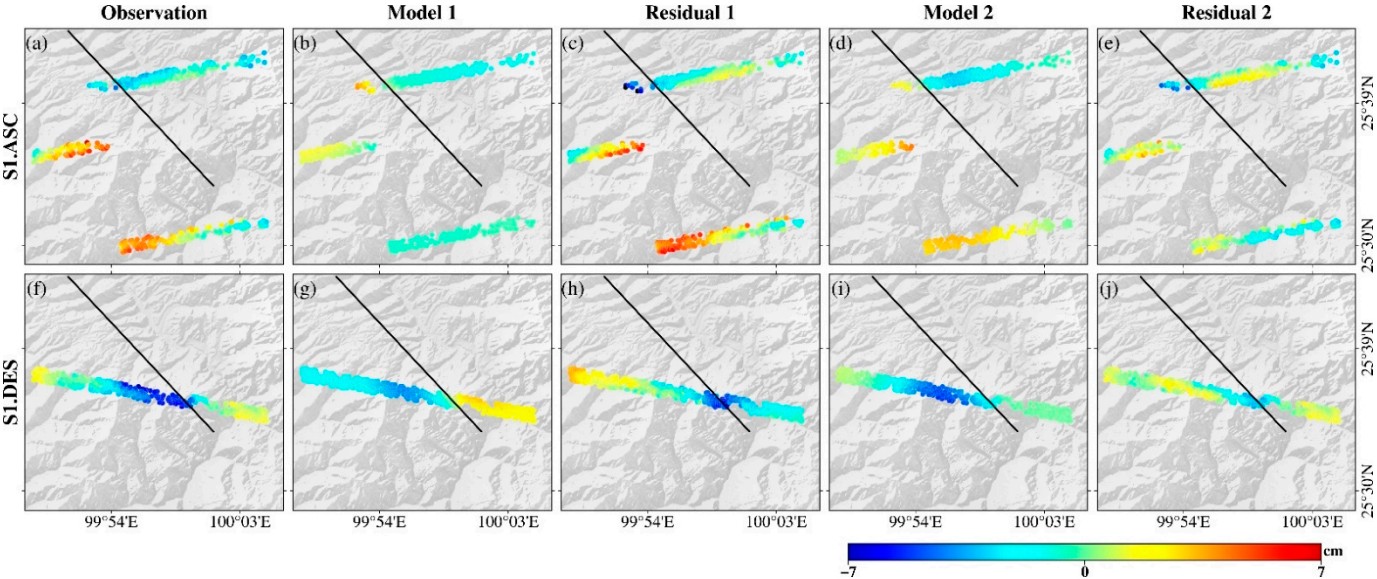

**Figure 9.** Observed and simulated BOI-derived azimuth coseismic deformation fields and residuals obtained by Model 1 and 2 from the ascending data (**upper row**) and descending data (**lower row**). (**a**,**f**) The observation fields. (**b**,**g**) The modelled displacement fields form Model 1. (**d**,**i**) The modelled displacement fields form Model 2. (**c**,**h**,**e**,**j**) The residuals from Model 1 and 2, respectively.

## 5. Discussion

### 5.1. Resolution Test of Two Groups of Models

Deriving coseismic displacement fields from more observed geometries is significant to constrain fault slip models. Adding the azimuth displacement in the near field obtained by BOI helps understand the dynamical process of seismogenic faults, especially the blind seismogenic faults. Although the area with azimuth displacements detected by BOI only takes 10% of the area of a single view SAR image, such measurement is very sensitive to near N-S deformation [12]. As Figure 4a shows, the BOI-derived azimuth displacements of the ascending orbit are coincident with the motion trend of the GPS horizontal displacement (positive displacements indicate along the direction of the satellite flight).

To test how well InSAR data can resolve the slip distribution in the fault plane, we perform checkerboard tests on InSAR datasets using only LOS deformation and the combination of LOS and BOI-derived azimuth deformation (Figure 10). We use geometries, dataset weights, and smoothing factors consistent with those of the fault model in linear inversion [40]. We construct two input models, one with five concave-convex bodies, each containing 4 × 4 sub-patches, and the other with nine concave-convex bodies with 3 × 3 or 2 × 2 sub-patches. The amount of slip on each sub-patch is set as 1 m. We calculate the displacements in the LOS direction and azimuth direction of the InSAR data based on the input model and add Gaussian noise to the result.

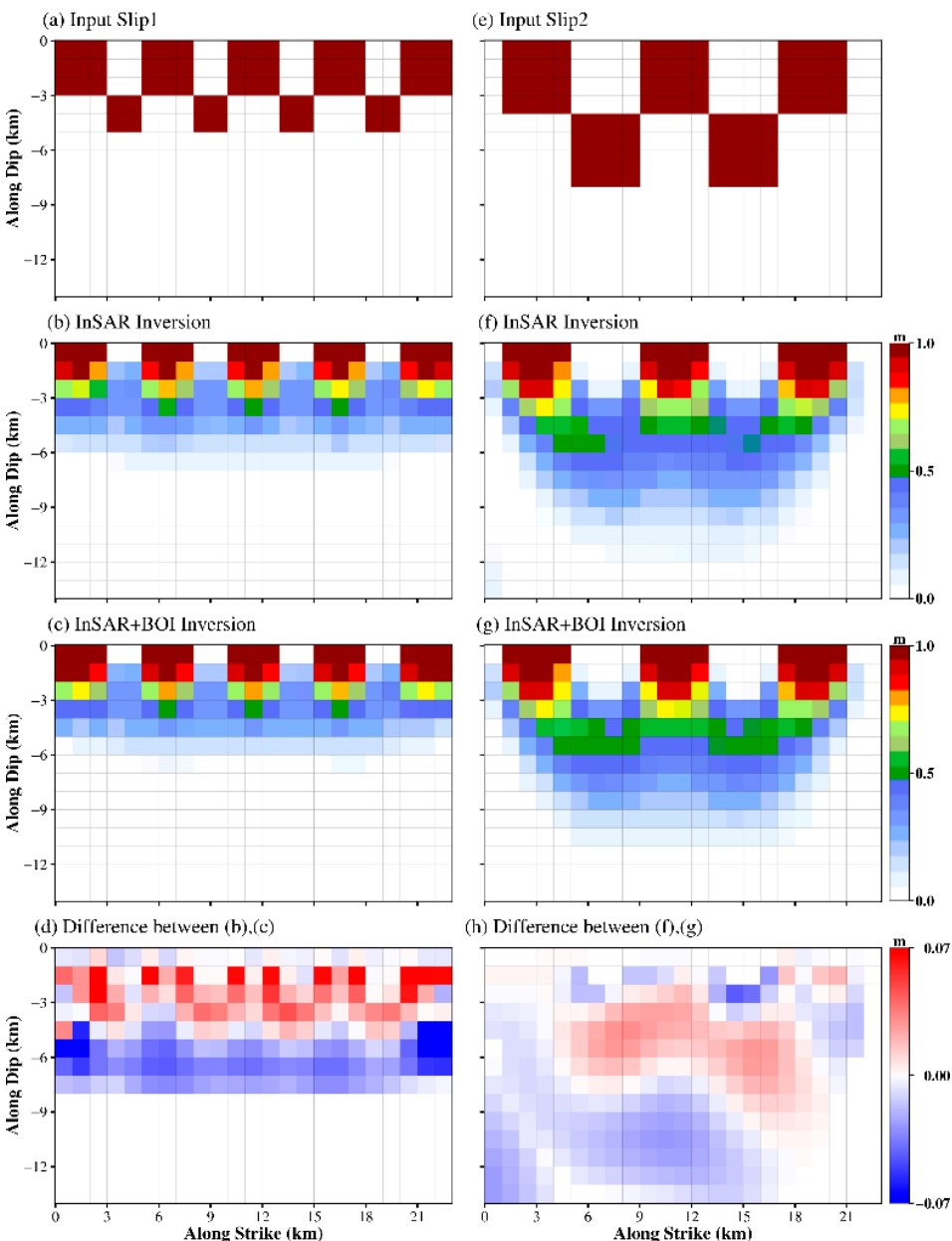

**Figure 10.** Model resolution for the checkerboard test. (**a**) Input slip model 1 and (**e**) input slip model 2 for calculating the synthetic displacements of InSAR data. (**b**,**f**) used InSAR LOS displacements. (**c**,**g**) combined InSAR LOS displacements and BOI azimuth displacements. (**d**,**h**) Difference between (**c**,**b**,**g**,**f**), respectively.

The slip distribution inverted from the InSAR LOS deformation (Figure 10b,f) and the joint inversion slip distribution (Figure 10c,g) show similar characteristics overall, as BOI

has fewer observations in the azimuth deformation than InSAR-derived LOS observations near the fault. In addition, in the joint inversion, the accuracy of InSAR data and BOI data is different. Furthermore, the BOI-derived azimuth observations have smaller weights than the LOS direction in the inversion. We calculate the difference between the model resolution test results and find that the simulated slip distribution based on the joint inversion of input slip1 has a maximum increase in slip volume of ~15% within ~5 km of the shallow part of the fault. In contrast, the slip volume decreases below 5 km of the fault depth (Figure 10d). Based on input slip2 the joint inversion of the slip distribution exhibits a maximum increase in slip volume of ~10% within 2 to 9 km of the fault depth (Figure 10h). The coseismic slip results in Figure 7b show that the slip is mainly concentrated in the middle of the fault approximately 6 km. The difference of slip distribution between the two model simulations shown by the checkerboard test is reflected in the slip increment within 7 km in the Northwestern part of the fault and 4~9 km in the Southeastern part of the fault (Figure 10h). The results show that the joint inversion model can better resolve the shallow slip of the fault than the coseismic slip model using only InSAR LOS displacement inversion, despite the limited azimuth constraints provided by BOI.

In addition, to evaluate the performance of the joint inversion model in resolving the amount of slip at different depths of the fault, we construct six input slip models (Figure 11). For the checkerboard model containing 4 × 4 sub-patches (Figure 11a–f), with the increasing number of concave-convex bodies, the joint inversion model gradually recovers more of the slip patches. Furthermore, if the concave-convex body is moved down 2 m (Figure 11g), the patches that can be recovered is significantly less than in Figure 11b. This shows that with the increase of fault depth, the resolution of the joint inversion model is constantly reduced, and reaches the lowest below ~11 km, which can be explained by the nearly vertical dip angle of the seismogenic fault [41]. As the sub-patch area of each concave-convex body increases, the simulated slip distribution can be resolved to more slip patches within the fault depth ~11 km (Figure 11j–l). In general, the results of the checkerboard tests show that combing InSAR LOS and BOI-derived azimuth displacements can produce a more refined and reliable slip model than using LOS alone.

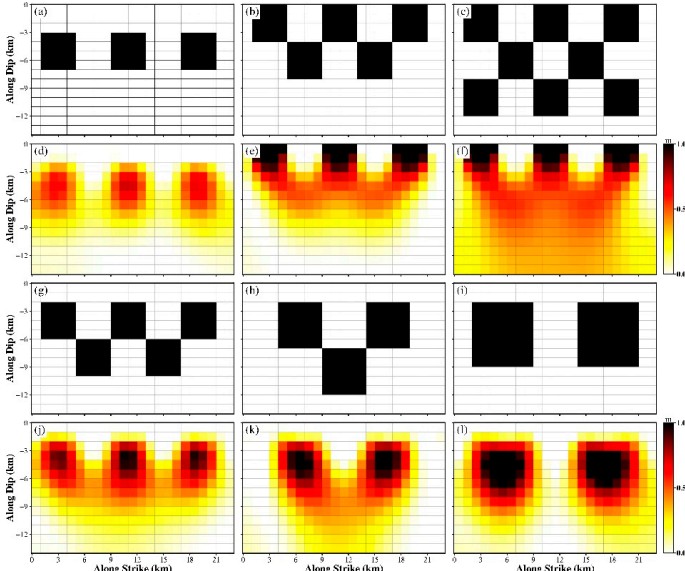

**Figure 11.** Checkerboard test results for joint inversion model based on different input models. (**a–c,g–i**) 4 × 4, 5 × 5, 7 × 7 sub-patches for each concave-convex body in the six input models, respectively. (**d–f,j–l**) are the corresponding joint LOS and BOI-derived azimuth displacements inversions of the slip distributions.

### 5.2. Comparison of Coseismic Slip Models

We compare the fault geometry, slip distribution, and magnitude of the coseismic slip model of the 2021 Yangbi earthquake obtained by three geodetic models [1,4,6]. We use the InSAR LOS displacements and the azimuth displacements detected by BOI for experiment. Compared with some studies using geodetic data (e.g., InSAR LOS deformation and GNSS data) [1,6], the deformation generated by BOI has more dense observations than GNSS, and contains more near-field azimuth displacement information. In addition, the joint inversion of these data produces significant constraints on the fault geometry and improves the resolution of the slip distribution of the seismogenic fault.

We also use a simplified fault model with a single fault strike and dip angle. Combining InSAR LOS displacements and GNSS data, S. Wang et al. [1] inverted the source parameters and got a fault dip of 80° and a SW tendency, which are the same as that obtained by K. Zhang et al. [6] who used only GNSS data. In this study, the inverted fault dip is 87°, consistent with that obtained by Y. Wang et al. [4] using InSAR LOS displacements inversion. K. Zhang et al. [6] obtained a deeper (4~12 km) coseismic slip distribution with a maximum slip of ~0.8 m using GNSS data inversion. S. Wang et al. [1] and Y. Wang et al. [4] obtained similar coseismic slip distribution to our model, which shows slips in the mid-Western region concentrating along the fault strike. However, the former obtained the maximum slip of 0.8 m, slightly smaller than that obtained by the latter, 0.94 m. The maximum slip obtained by our model is the largest, 1.1 m. In addition, most of the source parameters of our inversion are more consistent with those given by USGS and GCMT than the three models. Our geodetic moment magnitude is $1.66 \times 10^{18}$ Nm, corresponding to that of an $M_w$ 6.11 earthquake, which is closer to the results of USGS and GCMT ($M_w$ 6.10).

Our coseismic slip distribution model shows that the Yangbi earthquake exhibits a clear trend of rightward slip motion, which is consistent with the three geodetic models, indicating that our model is reliable. However, these models differ significantly in details, due to many factors, such as fault geometry, selection of data, kinematic assumptions, parameterization of the fault model, and smoothing factors. These details cannot identify which model is better. However, adding BOI-derived azimuth displacements will increase near-field data constraints to the fault model. This also helps to improve the understanding of the tectonic mechanisms of the seismogenic faults and to evaluate the contribution of the source model to the static rupture process of the Yangbi earthquake.

### 5.3. Coseismic Stress Changes and Potential Seismic Risk Assessment of WX-WSF and RRF

Coulomb stress changes often trigger secondary hazards in the surrounding areas after a major earthquake. To evaluate the hazard of the Yangbi earthquake, we use the inverted distributed slip model to calculate the Coulomb stress changes. The simplified formula is as follows:

$$\Delta CFS = \Delta \tau_s + \mu' \Delta \sigma_n \tag{3}$$

where $\Delta \tau_s$ and $\Delta \sigma_n$ are the shear and normal stress changes, respectively; $\mu'$ is the effective friction coefficient. Based on the Coulomb 3.3 open-source code from USGS, we calculate the stress released in the mainshock for a static frictional coefficient of 0.4 (Figure 12).

We project the relocated aftershock results onto the fault plane. We find most aftershocks occurred in the areas with increased coseismic stress, indicating that the Coulomb stress changes may play an important role in controlling the spatial distribution and evolution of aftershocks. This is similar to the aftershocks triggering in the 2021 Maduo $M_w$ 7.4 earthquake [42,43]. Such aftershocks can be caused by coseismic stress changes directly [44] or brittle creep in the fault zone [45]. If caused by brittle creep, the aftershock is considered to be the subsequent release of the stress increment generated by co-seismic slip on the seismogenic fault in an aseismical manner. Aseismic creeps become the dominant moment releasing mechanism [39,46]. In addition, the Coulomb stress at the shallow part of the seismogenic fault (1~3 km deep) increased significantly, indicating that this event may have triggered shallow sliding of the nearby faults [47]. We find a few aftershocks occurred in the region with reduced Coulomb stress. The possible explanations could be: (a) the

uncertainty in the aftershock depth used for repositioning, (b) aftershocks occurred near the main quake but not on the main fault plane, (c) afterslip triggered these aftershocks, and (d) aftershocks were triggered by increased Coulomb stress near the edge of the coseismic slip region.

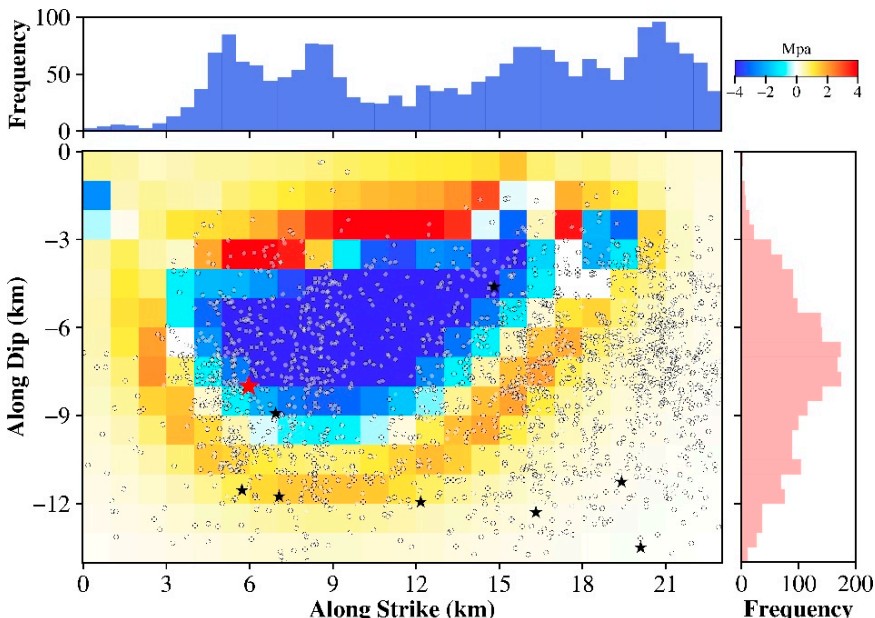

**Figure 12.** Relationships between aftershocks distribution and static Coulomb stress change. Coulomb stress change caused by coseismic slip is imaged in blue to red color scale. The red star shows the location of the mainshock, while the white circles denote the relocated aftershocks, and black stars are the $M_s > 4.0$ aftershocks. The distribution of aftershocks along the fault strike and tendency is shown as a histogram of the frequency distribution in blue and red.

The 2021 Yangbi earthquake occurred on a cryptic branch fault, apart from the WX-WSF in the Sichuan-Yunnan block, where four $M_w > 5$ earthquakes have occurred (Figure 1b). Studies show that moderate earthquakes usually change the stress field of the surrounding faults and increase the seismicity risk of the region [48]. We obtain the coseismic deformation field of the Yangbi 2017 $M_w$ 5.1 earthquake, but do not find any large deformation (Figure S6). Therefore, it is difficult to invert the source parameters from the coseismic deformation field and assess the impact of 2017 Yangbi event on the 2021 Yangbi event. There are different views on the seismic hazard of the RRF. Some believe that the RRF Zone becomes stable [21]. Some think that its North and South sections have a high strain rate accumulation closely related to the seismic activities [49].

We analyzed the effects of the 2021 Yangbi earthquake on the surrounding faults. We use the inverted seismogenic fault as the input model and assume that all known slip faults in the vicinity are upright (90° fault dip). We also assume that right-slip faults have a pure right-slip mechanism and left-slip faults have a pure left-slip mechanism. We calculate the coseismic Coulomb stress variation of the Southern end of the Weixi-Weishan fault (WX-WSF), the Red River fault (RRF1–4), the Yongsheng-Binchuan fault (YS-BCF), and the Lancang River fault (LCJF1–2) (Figure 13). The results show that in the Central part of the WX-WSF, the regional Coulomb stress in the Eastern part of the Yangbi Fault has increased by 0.14 Mpa, and the North-Central sub-region of the RRF also shows a Coulomb stress increment of ~0.05 Mpa (RRF3 and RRF4). In addition, there is a small increase in Coulomb stresses at the Southern end of the WX-WSF and the South-Central LCJF (LCJF1). The Yangbi event has increased the seismic risk in the surrounding areas, especially the WX-WSF and the RRF3. More importantly, the earthquake sequence of Yangbi, including the 2013 $M_w$ 5.4 earthquake, 2016 $M_w$ 5.0 earthquake, 2017 $M_w$ 5.1 earthquake

and $M_w$ 6.1 earthquake, all occurred on secondary blind faults of the WX-WSF. Future seismic risk in the region near the WX-WSF should still be of further concern.

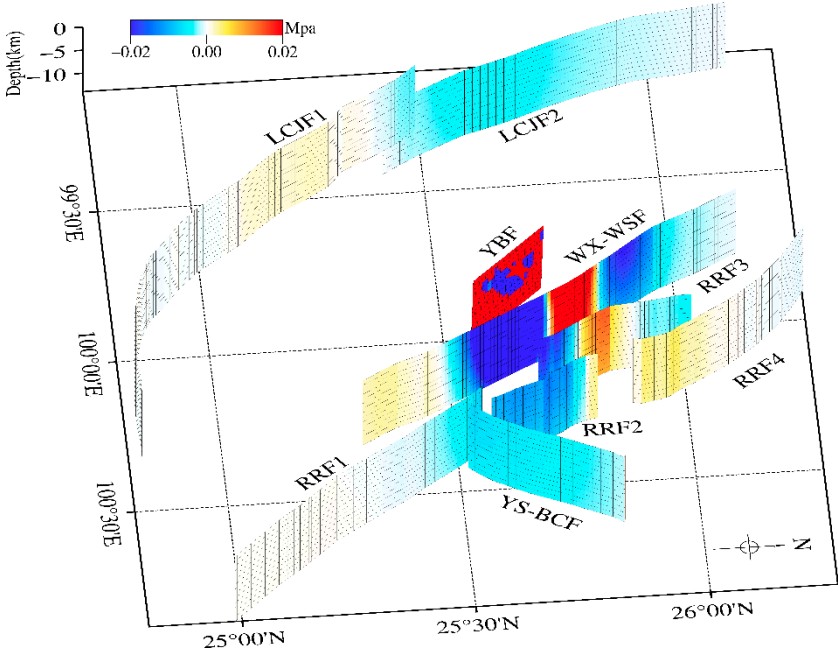

**Figure 13.** Coseismic Coulomb stress changes on the surrounding faults caused by the 2021 Yangbi earthquake. The magenta star denotes the location of the mainshock. YBF = Yangbi Fault, WX-WSF = Weixi-Weishan Fault; RRF 1–4 = Red River Fault 1–4; YS-BCF = Yongsheng-Binchuan Fault; LCJF 1–2 = Lancangjiang Fault 1–2.

## 6. Conclusions

We use D-InSAR and BOI techniques to acquire the coseismic LOS deformation and azimuth deformation fields of the 2021 Yangbi $M_w$ 6.1 earthquake in Yunnan China. The azimuth displacements in the burst overlap region demonstrate the ability of BOI for extending interferometry of moderate earthquakes. Our source fault model inverted from joint LOS and BOI-derived azimuth deformations shows that the Yangbi earthquake has a trend of right-slip motion, and the estimated geodetic moment magnitude is $1.66 \times 10^{18}$ Nm, corresponding to that of an $M_w$ 6.11 earthquake. The distribution of coseismic slip and aftershocks is complementary. The stress analysis suggests that the Yangbi event increased the Coulomb stress accumulation in the surrounding area, particularly in the central and southern part of the WWF and the northern part of the RRF. Therefore, geodetic observations and field geological investigations of unknown secondary fractures on these two faults are needed. In addition, further focus should be placed on the potential seismic risk in the southwest area of the WWF.

**Supplementary Materials:** The following supporting information can be downloaded at: https://www.mdpi.com/article/10.3390/rs14194804/s1, Figure S1: Interferogram network of the (a) ascending T99 and (b) descending T135. Figure S2: The 10 interferogram pairs of the ascending T99. Figure S3: The 10 interferogram pairs of the descending T135. Figure S4: Trade-off curve between the normalized misfit and model roughness. Figure S5: Fault slip distribution of the earthquake source model estimated by different smoothing factors of (a) 0.01, (b) 0.35, and (c) 0.05. Figure S6: Ascending coseismic interferograms of the 2017 Yangbi $M_w$ 5.1 earthquake.

**Author Contributions:** Conceptualization, H.L. and G.F.; data curation, H.L.; formal analysis, L.H., J.L., H.G. and X.W.; funding acquisition, H.L. and G.F.; methodology, H.L., G.F. and J.L.; resources, G.F.; software, H.L.; supervision, G.F., Y.W. (Yuedong Wang) and Q.A.; validation, G.F. and L.H.;

writing—original draft, H.L.; writing—review and editing, all authors. All authors have read and agreed to the published version of the manuscript.

**Funding:** This research was funded by the National Natural Science Foundation of China (No. 42174039) and the Fundamental Research Funds for the Central Universities of Central South University (No. 506021757).

**Data Availability Statement:** The Sentinel-1 SAR data used in this study are copyrighted by the European Space (http://scihub.copernicus.eu/dhus, accessed on 1 March2022). The moment tensor solution is from the Global Centroid Moment Tensor (http://www.globalcmt.org/CMTsearch.html, accessed on 1 March 2022) and the United States Geological Survey (https://www.usgs.gov/programs/earthquake-hazards/earthquakes, accessed on 1 March 2022). The GBIS software was obtained from https://comet.nerc.ac.uk/gbis/ (accessed on 1 March 2022).

**Acknowledgments:** The authors would like to thank the European Space Agency (ESA) for providing free Sentinel-1 data, and three anonymous reviewers for their thoughtful and constructive comments that greatly improved this manuscript. The figures in this study were generated by the Generic Mapping Tools [50].

**Conflicts of Interest:** The authors declare no conflict of interest.

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
