# Peer review of "An Improved Source Model of the 2021 Mw 6.1 Yangbi Earthquake (Southwest China) Based on InSAR and BOI Datasets"

_remotesensing, doi:10.3390/rs14194804_

Round 1

Reviewer 1 Report

In this manuscript, the InSAR-derived LOS and BOI-derived azimuth displacements from Sentinel-1 data are combined to constraint the source model of the 2021 Yangbi earthquake. The inversion results verified the combined data and the potential seismic risks on the Weixi-Weishan and the Red River faults were predicted.

There are some flaws to be modified:

1. The E-W and vertical components of the deformation of the 2021 Yangbi event is shown in Fig. 3 by ignoring the N-S displacement. It is suggested to give the E-W and vertical components using the BOI-derived azimuth displacement in Fig.4 in comparison to Fig.3.

2. “A modified Goldstein filtering method is used to filter the interferograms [25]”. The filter [25] was presented in 2008. Why are the latest denoising methods[R1-R2] not applied? Is the filter [25] enough for Sentinel-1 data?

[R1] Xu, H. et al. A Nonlocal Noise Reduction Method Based on Fringe Frequency Compensation for SAR Interferogram. IEEE Journal of Selected Topics in Applied Earth Observations and Remote Sensing, 2021, 14: 9756-9767.

[R2] H. Ansari, F. De Zan and R. Bamler, "Efficient Phase Estimation for Interferogram Stacks," in IEEE Transactions on Geoscience and Remote Sensing, vol. 56, no. 7, pp. 4109-4125, July 2018.

3 Slight improvement is provided by combining the azimuth deformations in this manuscript. Is it because the overlapped belts, providing azimuth deformations, occupy small part of the area?

4. Line 69 of page 2, the explanation of abbreviation BOI should be given when it appears firstly on line 62 of the same page.

5. The colorbar should be added to the third row of Fig.10 and the second row of Fig.11.

Reviewer 2 Report

This study demonstrates the utility of burst overlap interferometry (BOI) in delineating the slip distribution of the 2021 Yangi earthquake, southwest China. This study is well motivated and generally organized, but I see some awkward expressions in the manuscript. 

I believe the manuscript will be improved by addressing my comments, as enumerated below. The main comment is that the authors can better demonstrate the utility of BOI because BOI is obviously powerful in observing the surface deformation and thus delineating the slip distribution of the earthquake in this case. After all, the earthquake deformation results from a strike slip on a quasi-north-south striking fault, generating substantial north-south displacements. 

1. I appreciate the authors' effort in demonstrating the utility of BOI in delineating the slip distribution of an earthquake, but I think the authors can do better. The authors compare the inverted slip distribution with and without BOI data and are not very successful in demonstrating the statistical significance of adding BOI. It is not surprising, given that BOI displacements constitute only a tiny portion of the dataset. In this case, I suggest the BOI data and the expected BOI observation from the model without BOI observations. I suspect that they are significantly different. If so, this discrepancy means that the model without BOI is at least not perfect because it cannot explain a new dataset. 

As a next step, the authors might want to construct a model by including BOI data but removing some of the InSAR measurements. Then you can compare the removed InSAR data and the expected InSAR observations from the model. I think they will not be too different. The authors will be able to demonstrate the utility of BOI by doing these experiments, for example. 

2. Table 3 and Figure 6 indicate that the fault responsible for the earthquake is long and narrow. The scaling law of earthquakes expects that the width of M<7.5 earthquakes, which do not rupture the whole seismogenic zone, is about half its length, but the authors' model favors a much narrower fault with its width only 1/6 of its length. The authors need to interpret this finding. Is such a narrow fault because of a thin seismogenic zone? Are there some other reasons to explain this observation? 

3. Related to the comment above, is the slip distribution the authors obtain because of an assumption of homogeneous halfspace? Previous studies suggest that ignoring the crustal layering underestimates the depth of the fault slip (e.g., Manconi et al., 2007, 10.1111/j.1365-246X.2007.03449.x). 

4. The authors try to understand the aftershocks by static stress perturbations by the mainshock (Section 5.3). It is a reasonable approach, but not all aftershocks must occur where the Coulomb stress change is positive. Dynamic stress changes, which are not considered in this study, play a substantial role in triggered seismicity (e.g., Brodsky and van der Elst, 2014, 10.1146/annurev-earth-060313-054648). 

5. Line 35 Ms 6.4: The authors use the surface-wave magnitude here, but the moment magnitude in the title. I recommend making the magnitude scale consistent. You should use moment magnitude or surface-wave magnitude consistently throughout the manuscript. 

6. Table 1: Citations of Y. Wang et al., B. Zhang et al., K. Zhang et al., and S. Wang et al. necessary. 

7. Line 59-60 ...an earthquake has an N-S trending: The north-south displacement dominates only in the case of strike-slip faulting. 

8. Line 140: I did not understand what the amplitude thresholds mean. Is it related to the amplitude change over time? How is this metric calculated?

9. Line 292: Employing the second-order Laplace smoothing constraint is fine, but the weight of this constraint and how the authors choose this weight need to be described. 

10. Figure 8: Are residuals drawn with the same color scale as models? Probably not. Also, the authors should show the observation next to the model for comparison. 

Reviewer 3 Report

Congratulations on your interesting and instructive paper.

You investigated on the base of DInSAR remote sensing images the parameters and mechanism of the Yangbi 2021 earthquake in China.

In the paper, you propose and discuss a technique named Burst Overlap Interferometry to derive seismic shifts in north south direction more precisely than it is applicable by other methods. These shift measurements serve as input for seismic models.

Your paper is well structured and scientifically sound. The scientific background of the discussed algorithm is presented in detail. All conclusions drawn in the text all well supported either by measurement results or by theoretical considerations.

English usage and style is correct. The number and choice of cited references is adequate. Basic works and most recent papers at the area of research are adequately considered. Beyond minor text editing, I recommend the publication of your paper in its present form.

My only comments to be considered in the final version of the paper are:

page 2, line 45-57: Please define close to Table 1 the meaning of the acronyms GCMT and USGS. I found the definition of USGS two pages too late while GCMT is not defined anywhere in the paper.

page 5, line 161-162 (just a remark, no action item w.r.t text editing): You eliminated the influence of atmospheric delay on the measurement results by spatial and temporal filtering. In contrast, several recent scientific studies at the area of SAR and InSAR presented encouraging results from subtracting the GNSS measurement based or model based values of the atmospheric delay prior to InSAR processing. Are you aware of these improvements?

page 6, line 180-182: "where d_LOS is the LOS deformation [...], respectively" Please define those variables which are already used in eq. (1) closer to eq. (1) in the text. I suggest to shift the entire phrase from line 180-182 to line175.

page 6, line 195: typo: "as slide as" Obviously, you mean "as wide as"

page 12, line 370: wrong word: "We difference" What do you mean by this? "We calculated the difference" or "We compared" or ...?

Round 2

Reviewer 2 Report

I am satisfied with the authors' response to my comments and I am happy to recommend acceptance of this manuscript.